# Describing the Pollen Content in the Gastrointestinal Tract of *Vespa velutina* Larvae

**DOI:** 10.3390/ani13193038

**Published:** 2023-09-27

**Authors:** Ana Diéguez-Antón, Olga Escuredo, Laura Meno, María Carmen Seijo, María Shantal Rodríguez-Flores

**Affiliations:** Department of Plant Biology and Soil Sciences, University of Vigo, 32004 Ourense, Spain; ana.dieguez.anton@uvigo.gal (A.D.-A.); oescuredo@uvigo.es (O.E.); laura.meno@uvigo.es (L.M.); mcoello@uvigo.es (M.C.S.)

**Keywords:** *Vespa velutina*, gastrointestinal content, pollen grains, larvae

## Abstract

**Simple Summary:**

The yellow-legged hornet is an invasive species from southeast Asia that has turned the European beekeeping sector upside down. The spread of this species has been advancing in recent years, and today, several European countries are threatened by *Vespa velutina*. The need to study its behavior is urgent given the increasingly evident economic and environmental impacts. In this regard, there is little information about the feeding habits and the resources it uses during the life cycle. Like other Hymenoptera, hornets require carbohydrates and proteins as their primary nutrients. Sugary secretions such as floral nectar, honeydew, or fruit juices are the main sources of carbohydrates but the protein intake is provided by the consumption of a diverse diet of insects such as the honey bee. There is scarce information on the presence of pollen grains in the gastrointestinal content of larvae other than secondary contamination from hunting. This content could represent the surrounding flora of its habitat that is used as a resource. Therefore, the objective of this study was to describe the main pollen types present in the gastrointestinal system of larvae taken from *V. velutina* nests.

**Abstract:**

*Vespa velutina* is an invasive species that exhibits flexible social behavior, which may have contributed to its introduction in several European countries. It is important to understand its behavior in order to combat the effects of its introduction in different areas. This implies knowing the resources that it uses during its biological cycle. Hornets require protein resources taken from insects and organic matter as well as carbohydrates as an energy source to fly and also to forage for food and nest-building materials. The gastrointestinal tract of adults and larvae contains a wide variety of pollen types. The identification of this pollen in larvae collected from nests could offer information about the plant species that *V. velutina* visits as a foraging place. The main objective of this research was to study the pollen content in the gastrointestinal tract of larvae. Patterns of pollen content and pollen diversity were established according to the nest type, altitude, season, and location in the nest comb. The abundance of pollen types such as *Eucalyptus*, *Castanea*, *Foeniculum vulgare*, *Hedera helix*, *Taraxacum officinale*, *Echium,* or *Cytisus* pollen type stands out in many of the samples.

## 1. Introduction

The Asian hornet *Vespa velutina nigrithorax* du Buysson, 1905 (Hymenoptera: Vespidae), is a social hornet native to southeast Asia and naturally distributed in southern China, India, Indochina, and Indonesia [1,2]. The species was accidentally introduced in some areas of South Korea and Japan as well as in Europe, where the first specimens were detected in France almost 20 years ago. The behavior of the species, the ability to adapt to new environments, and the voracious predatory activity facilitate an agile introduction in some European countries, posing a threat to human activities and the conservation of the natural environment and biodiversity. Therefore, since 2016, this species has been included in the list of invasive alien species of concern for the Union (EU Regulation 1141/2016), which requires the development of monitoring plans and measures to limit its spread as well as control and containment strategies [3]. In the case of Galicia (northwest Spain), the first sightings were made in 2012 and 10 years later, the number of colonies detected by the surveillance programs is over 30,000 [4]. 

The social behavior of the hornet colony can be understood as a superorganism in which collecting and regulating nutrients is achieved through the collective action of workers [5]. The organization of these social insects is one of the factors that contribute to their colonization. In particular, it allows large colonies to form and remain active throughout the cycle.

The biological cycle is annual, usually beginning in February when the weather conditions are favorable for mated queens to fly, to suck nectar and other sugary secretions, and to perform embryo nest-building activities [6]. In the first stage, the queen begins to build a small and fragile nest with a comb for egg laying. The first hornets emerge after about 50 days [7] and are often smaller than the hornets that emerge in the summer and fall. The smaller cell size and probably poorer nutrition compared to later-born workers account for these differences. As the cycle progresses, the embryo nest becomes larger; it may continue to grow in the same location or the colony may move to another location with the construction of a new nest [8]. Large colonies require large amounts of food. Sugary secretions are important for energy but protein resources are taken by predation on other insects or by cutting up pieces of organic matter from animals. When the colony reaches its maximum size and population, the time for larval development and metamorphosis is reduced to 29 days and this is the time when the most damage occurs in apiaries [4,7,8]. The period of nest decline begins when the colony produces mostly males and future queens (mainly in autumn) which will be the new foundresses. This stage lasts until when the unfavorable environmental conditions limit the flight of the workers. In the specific case of Galicia, this moment can last until February. The nest is then gradually depopulated and finally, it is completely abandoned. In the meantime, the new queens choose a sheltered place to spend the winter. 

One of the most important social behaviors for social insects within the colony is trophallaxis [6,9]. Trophallaxis can occur between adults or between adults and larvae. To initiate adult–larvae trophallaxis, a worker must locate the resource in the environment. The forager must subdue the prey, and, if it is too large to be carried back in one piece, it must fragment the prey before the hornet returns. Once inside the nest, the worker approaches a larva, usually observed when the adult inserts its head into a brood cell. The adult hornet stimulates the larva by touching its mouth parts and then, the larva secrete the oral exudate (saliva) which is transferred to the adult’s gastric tract [10,11,12]. Larval saliva in the trophallaxis between larvae and adults and controls the social life of hornets and also plays a crucial role in the control of ecological and instinctive behaviors. Larval saliva is physically essential to the life of hornets and a lack of saliva or a lack of trophallaxis results in the death of individuals and the decline of the colony [12,13]. Each wasp species has a peculiar amino acid composition of larval saliva and it has been probed that the larval saliva of *V. velutina* has a high ratio of free amino acids and a high proportion of proline [14]. These are thought to correlate with the daily flight distance and the size of the domain of *Vespa* species [12,15].

Salivation by the larvae is stimulated at the time when the workers are feeding the carnivorous larvae with different protein sources, with the salivation acting as a reward for the workers. In parallel, workers also feed on a diet rich in carbohydrates found in sugary liquids such as plant nectar [16]. This provides them with the necessary energy to make long flights and search for prey. The attraction to these resources is related to the recognition of odor signals that can be detected over long distances [17,18]. *V. velutina* also recognizes chemical signals from apiaries being attracted to honey and bee pheromones. This hornet is a species that finds apiaries attractive because they provide a concentrated source of protein. Workers generally hover over the entrance to the hive and capture flying honey bees, some of which are laden with pollen and/or nectar [4].

Many social insects visit plants to collect pollen grains as an important source of protein and amino acids. In the case of the Asian hornet, the use of pollen grains as a resource has not been described. This species does not have structures that allow it to collect and transfer pollen grains from flowers to the colonies, but it has been observed that significant amounts of partially digested pollen can be found in the intestinal tract. The importance of plant species as resources for the yellow-legged hornet has not been investigated beyond the fact that plants are places to find prey and often contain pollen adhered to the body [19,20]. The presence of pollen in the gastrointestinal contents of larvae of this species could indicate the main plants of interest for *V. velutina* as foraging places throughout its life cycle. For this reason, this study analyzed the content and diversity of pollen grains found in larvae of Asian hornets collected from nests located in different points of Galicia. Even though the coastal areas continue to be the ones with the highest number of nests [8], the latest data from the Xunta de Galicia, through the *Vespa velutina* Hornet Surveillance and Control Program, have registered nests throughout the Galician territory. Therefore, this species is currently invading any area regardless of altitude. This variable was studied in order to identify diversity and pollen content at different altitudes. This information aims to contribute to the knowledge on *V. velutina* and, in particular, which floral resources are important in the habits of this species. 

## 2. Materials and Methods

### 2.1. Study Area

Galicia is located in the extreme northwest of the Iberian peninsula between two climatic environments: the Atlantic and the continental plateau. This gives it a climatic diversity that determines the peculiarities of its vegetation halfway between the Eurosiberian and the Mediterranean regions. The topography is characterized by the existence of mountain ranges with altitudes higher than 1600 m located 200 km from the coast, in the southeast, while in the north, the elevation is approximately 500 m and the coast is 20 km away. 

### 2.2. Data Collection

For this study, a total of 56 nests from highly invaded areas were collected. The nests came from different locations both in coastal and inland areas. The pollen content of the digestive tract of a total of 675 larvae extracted from nests was studied. Table 1 shows a description of the nests from which the larval samples were obtained.

Nests from different altitudes above sea level were also sampled. The altitude of the nests was grouped into four types: ≤100 (from 2 m to 100 m above sea level); 100–200 (from 106 to 170 m above sea level); 200–300 (from 230 to 285 m above sea level); and >300 (from 350 to 600 m above sea level).

Nests were collected in different periods of the year (spring, summer, autumn, and winter). They were also classified as secondary nests or embryo nests according to their characteristics. For each nest, the combs with larvae were separated and four larvae were selected in each comb for the study. The number of combs used was 122.

Regarding the location, the nests were detected both in inhabited areas (buildings or sites related to human activity, especially embryo nests) and in forest areas (trees and bushes), which provide them with a hiding place in which they are easily camouflaged (mainly secondary nests).

### 2.3. Gastrointestinal Dissection

The larvae involved in this study were collected from unsealed cells, following a predefined pattern according to their position in the comb, to sample larvae from distant positions, as far as possible. Once the sample was taken, each larva was conveniently identified and preserved separately in the freezer at −20 °C until the study of the gut content was carried out.

The gut contents of each larva were removed by dissection under a binocular microscope. Each larva was opened longitudinally following the direction of the digestive tract. Once opened, the entire digestive tube was removed with the aid of small forceps. A scalpel was used for insect dissection. Each digestive tube was placed in a Falcon tube and was mixed with 5 mL of distilled water and 2.5 mL of ethanol (80%). This solution was smashed and homogenized as best as possible. Later it was filtered using a perforated plate sieve with a pore size of 100 μm.

### 2.4. Slide Preparation and Identification of Pollen Content

The slides were prepared taking two drops of 100 µL of the solution described before and depositing both separately over a slide. A cover slide with a drop of glycerol-gelatine and fuchsine was added to each drop. The identification of pollen grains was carried out using an optical microscope Olympus at 400× or 1000× when necessary. The pollen grains in each drop were identified and the number was estimated considering the size of the microscope field of view, the surface used for counting, and the volume of the sample. The results were expressed as pollen grains/10 μL. The identification was performed by a specialized analyst in palynology. The pollen types were named according to the level of identification. When the morphological features of the pollen grain allowed for full identification, they were given the name of the corresponding plant species. The genus name was used when only a similar genus morphology was found. Finally, the pollen type (t.) denomination was used for pollen grains that only presented similar morphology for some genera and even botanical families.

### 2.5. Statistical Analysis 

The statistical analyses were conducted using Statgraphics Centurion 17.0 for Windows (Statgraphics Technologies, Inc., The Plains, VA, USA). The pollen content was compared using intervals based on Fisher’s Least Significant Difference (LSD) procedure. These intervals were constructed in such a way that if two means are equal, their intervals will overlap 95.0% of the time. Any pair of intervals that do not overlap vertically corresponds to a pair of means that have a statistically significant difference. 

A principal component analysis (PCA) was carried out to better understand the existing correlations between the variables studied and to establish hypotheses about the interrelationships between them. The PCA was executed to reduce the dimensionality of the data matrix and establish significant connections between the altitude, season, month, number of combs, number of pollen types per nest (NPT nest), number of pollen types per comb (NPT comb), and number of pollen types per altitude (NPT per altitude) variables.

## 3. Results

### 3.1. Principal Elements Identified in the Gastrointestinal Content

Various structures were found in the gut content of *V. velutina* larvae as a result of their diverse diet. As shown in Figure 1, pollen grains stained with fuchsin can be observed (PT) in all the studied larva samples (n = 675). The amount of pollen content found in the gut varied from less than 10 pollen grains/10 μL (4.2% of samples) to a maximum of more than 500 pollen grains/10 μL (2.3% of samples). Most of them (75.9% of larvae) presented between 10 and 200 pollen grains/10 μL. In addition, small fragment structures originating from the bodies of other insects were seen (RS). 

The obtained slides were rich in different types of elements, but the presence of pollen grains was common in all of them.

### 3.2. Principal Plant Families and Pollen Types

This study focused especially on the pollen content found in the larvae of *V. velutina*. 

Thirty-nine families represented the pollen types found in the digestive contents. The most abundant were Asteraceae, Fabaceae, Fagaceae, and Rosaceae. A total of 68 pollen types were identified. Most of these families are part of the surrounding wild vegetation but it is worth mentioning the appearance of species corresponding to crops such as *Eriobotrya japonica*, *Citrus,* or *Laurus nobilis*. The main pollen types and the families to which they belong can be seen in Table 2.

Despite this pollen diversity, the frequency of these pollen types was low for most of the identified pollen types. However, *Castanea*, *Taraxacum officinale* t., *Eucalyptus*, *Genista* t, *Erica*, *Hedera helix*, *Foeniculum vulgare* t., and *Echium* pollen types were present in more than 50% of the samples (Figure 2).

*Castanea* pollen was the most representative, being present in 86.2% of the samples, followed by *Taraxacum officinale* t. (82.8%) and *Eucalyptus* (82.8%). Regarding the values for each pollen type in larva samples, *Hedera helix* and *Eucalyptus* stand out. Both reached maximum values higher than 90% of the total pollen grains identified. However, *Castanea sativa,* despite being very frequent in samples, does not exceed 20%. Other pollen types such as *Cytisus*, *Taraxacum officinale*, *Echium*, *Erica,* and *Foeniculum vulgare* also had low values.

### 3.3. Number of Pollen Types and Diversity According to the Nest Type and Altitude

The studied embryo nests were collected from March to the beginning of June and had a maximum of three combs. Secondary nests often had more than three combs and were collected from the months of May to December. Some secondary nests were collected in February since they were still active. 

If the type of nests is considered, the embryo nests had a lower diversity (different types found in all the samples) than the secondary ones. A total of 47 different pollen types were found in the embryo nests while in the secondary nests a total of 67 pollen types were identified. Overall, 43 of them were common in both types of nests (Table 3). 

Regarding the mean number of pollen types found, a similar pollen type content was detected in both types of nests since an average content of pollen types (NT) of 9.4 was identified in the embryo nests and an average NT of 8.2 occurred in secondary nests.

The altitude of the nests influenced the diversity of pollen types (S) but not the pollen content (NT). The larvae analyzed from nests located at low altitudes (≤100) presented a greater diversity (S) of pollen types than those located at higher altitudes. If we consider the type of nest, this diversity was even higher in secondary nests (S) = 54 versus 46 of the embryo nests. Regarding the content of pollen types found, it was similar in all nests regardless of altitude, with no significant differences (Table 3).

### 3.4. Pollen Patterns across the Life Cycle

A variable pollen content has been identified in the gastrointestinal contents of the larvae throughout the life cycle. The month of March presented the highest average number of pollen types (Figure 3a). Until the month of June, the number of types of pollen detected in the gastrointestinal content decreased. This content increases again in the larvae analyzed in the months of July and September. Finally, the average content of pollen types in December and January decreased, being significantly lower than the rest. 

If we consider the number of observations of different pollen types or diversity (*S*), in November, a total of 45 different pollen types were identified while the months in which a lower diversity of pollen types were January and February, with 17 and 10 different pollen types, respectively (Figure 3b).

Figure 3b shows a plot of the main pollen types used in the life cycle. *Eucalyptus-* and *Hedera-type* pollens were the most abundant in larval samples. *Eucalyptus* presented a mean maximum value of 83% in March and *Hedera* had a mean maximum value of 39.6% in October and 43.6% in January. Other pollen types such as *Castanea* and *Foeniculum vulgare* t. appeared in higher percentages starting in April. The content in *Calendula* stands out from March to May and the presence of *Rubus* increases in June. This could show that these are resources highly related to this species, regardless of whether it can be acquired by contamination in the intake of other insects such as *Apis mellifera*. 

### 3.5. Pollen Patterns Depending on the Comb

No significant differences were observed for pollen richness depending on the comb studied. However, the oldest combs presented a greater diversity whereas the medium combs had a greater quantity of pollen types. In this way, the newest had the least amount of pollen types and diversity (Table 4).

Larvae from comb number 9 contained the highest average number of pollen types (9.4 NT). It is also worth noting comb number 1 (oldest comb) presented the larvae with the higher pollen diversity (S) = 51, followed by comb 2 and 3. In addition, larvae from comb 8 presented up to 22 pollen types in some of the samples studied.

### 3.6. Relationship between the Content Pollen and the Features of the Nest

Seeking to summarize and relate each of the variables studied a principal component analysis was carried out. The purpose of the analysis was to obtain a reduced number of linear combinations of the altitude, season, month, number of combs, number of pollen types per nest (NPT nest), number of pollen types per comb (NPT comb), and number of pollen types per altitude (NPT per altitude) variables that explain the greatest variability in the data. In this case, three components have been extracted since they had eigenvalues greater than or equal to 1.0. Together, they explained 78.0% of the variability in the original data (Figure 4).

The graph in Figure 4 shows the first two principal components. In each of the lines, the variables studied are shown and their proximity indicates a correlation between them. Considering the weight of each variable in each component are considered, the qualitative variables, number of combs, month, and season, were the ones that had the greatest weight for component 1 whereas for component 2 the quantitative variables NPT by nest, NPT by comb type; NPT by altitude had a greater weight.

Regarding the correlation between the variables, it should be noted that the more advanced the nest is, the lower the NPT it has, regardless of the type of nest. Altitude is another factor that affects pollen content, thus, at higher altitudes the larvae of each nest will have a lower number of pollen types.

## 4. Discussion

The feeding of the offspring is based mainly on a diet rich in protein. Like other social hornet species, the Asian hornet is a predator of other insects. This resource will mainly provide them with proteins that they use to feed the larvae. However, this species collects both liquid and solid food in the field. Liquids are mainly used to satisfy the carbohydrate intake and are also distributed to adult nestmates [16].

Nectar and pollen are nutritionally superior to other plant materials and are the main source of carbohydrates for many hymenopterans. *V. velutina* requires carbohydrates, as a key source of energy, for activities such as flight [21]. In fact, hornets are frequently observed visiting flowers as a foraging place [17,19,20]. 

In the present study, both remains of other insects and different pollen types have been observed in the gastrointestinal contents of the larvae. Some of the pollen types were observed to have their exine partially digested. This could indicate an indirect contribution of protein to larval feeding, as described in the case of the honey bee [22]. Insect diet identification of both *V. velutina* adults and larvae is currently being studied using DNA metabarcoding as a method that allows nest triangulation and destruction combined with active surveillance and control [21,23]. However, the role this pollen has in the nutrition of *V. velutina* is not clear, therefore, further research would be needed.

The adult workers are responsible for feeding the larvae [24,25]. These hunt different species of insects in flight, such as different species of bees, other wasps, dipterons, and even arachnids [23]. The workers process a protein meatball that is taken back to the nest for feeding the larvae but do not consume it directly. As the colony grows, the larvae’s demand for protein increases, especially in late summer and during the fall months. The larvae also collaborate in feeding the colony by means of a reciprocal exchange of food (trophalaxy). In the particular case of the larvae, they reward the workers using a highly energetic oral secretion, rich in sugars, proteins, and free amino acids, which creates a dependency on the worker and ensures that the larvae are fed regularly [13,24]. This could justify the presence of pollen in the gastrointestinal content of the larvae. In addition, it is important to highlight that most of the insects hunted by *V. velutina* contribute to pollination, thus having an impact on this biological process [19]. Therefore, the probability of this pollen deriving from the body of hunted insects (protein meatball) is very high.

Plant resources can also be used to build the nest. *V. velutina* is a social species of hornet that forms colonies. Each colony lives in a nest that can produce an average of 13,000 individuals and up to 23,000 with nine combs [8]. These large nests have been found particularly in the crown of trees such as *Eucalyptus globulus* Labill. or other perennial trees [26]. Two cultivated eucalyptus species are well distributed in Galicia, *E. globulus,* and *E. camaldulensis*, of which the first is the most significant. *E. globulus* abounds along the entire coastal margin as a consequence of the intensity of reforestation with this fast-growing species. This species reaches a large size where it is found forming monospecific forests. The most stable flowering takes place during the months of November and March, achieving a high yield in nectar secretion [27]. *E. camaldulensis* adapts better to higher altitudes and a more continental climate, which is why it is more abundant than *E. globulus* in the Galician interior. It blooms between the months of May and July, producing abundant pollen and nectar. This contribution of varied resources could make this a species of interest for *V. velutina*. being used as a place of protection, as a food resource, and as a resource for the elaboration of the nest. That is why it has been identified in the gastrointestinal content of the larvae throughout their entire life cycle.

However, a high pollen diversity was identified in the larvae. The highest content of types of pollen stood out at the beginning of spring, coinciding with the emergence of the foundress queens who seek carbohydrates from the nectar of plants that flower in spring as the first source of food [28]. Some of the pollen types are part of the vegetation of beekeeping interest, such as *Castanea, Taraxacum officinale,* or *Echium.* In the case of *Castanea*, this beekeeping interest stands out, especially throughout the Galician territory. The high frequency found in the samples could be due to additional contributions of this pollen in the bee prey that are hunted by the workers. Although it is true that these plants are of interest to most pollinating insects, *V. velutina* is a voracious predator of honey bees, especially during the summer and early autumn months, causing serious impacts on the apiaries. Hornets capture their prey in flight or sit in the entrance of the hive, discarding wings, legs, and abdomen and collecting only the thorax, rich in muscle protein. That is why a higher content of pollen types in the studied larva samples has also been found during the months of higher pressure in the hive (July and September). This content could be the result of contamination from honey bees or even other pollinating insects. However, a high pollen diversity has been observed during October and November, which could mean that hornets have had to interact with a wider variety of plants in search of carbohydrates and insects in order to have the same nutritional input as the previous month to feed their larvae.

The choice of resources used is influenced by the location of the nest. The shape and size of the nest and therefore the number of offspring produced, is influenced by factors such as altitude [26,29]. In lower altitude areas, a greater diversity of species was identified in the gastrointestinal content of the larvae studied. However, the workers incorporate a similar number of pollen types into the diet of the larvae at any altitude. This could mean a constant feeding pattern independent of both the nest types and the altitude at which the nest is located. This also implies the species is a very successful invader regardless of altitude conditions opposite to what was initially thought [26]. 

If the structure of the nest is considered, the pollen content did not present differences by the comb, even differentiating between the embryo and secondary nests. However, the larvae from the oldest combs presented a greater diversity of pollen types. The greater diversity among the larvae of the first combs could be influenced by the content of the larvae of the embryo nests. The first combs in the embryo nests are formed during the spring when there is a higher content of pollen types and therefore the larvae showed a greater plant diversity in their gastrointestinal tract during the season. Further research could improve the understanding of plant foraging of interest for the hornet.

## 5. Conclusions

The gastrointestinal content of *V. velutina* larvae has confirmed the presence of both insect remains and pollen types. In this study, the main pollen types and the content and diversity of these pollen types were identified. The main pollen types are those belonging to the flora surrounding the *V. velutina* habitat. This content belongs to the flora that the hornets use as a resource to feed themselves, feed their progeny, and build nests. These resources have varied depending on the altitude at which the nest was located, in such a way that a greater pollen diversity has been seen in the larvae that belong to lower altitudes. A decreasing pollen content was identified as the larvae from the last months of the year were collected. The pollen contents stood out when the foundress queens emerged (spring) and in the months of greatest pressure in the hive (July and September). Within the same nest, the larvae have presented the same pollen content; however, the diversity of pollen types is greater in the older combs. The results of this study show that *V. velutina* used certain plant resources as foraging places (for nectar and animal protein). The identification of pollen from the gastrointestinal content of larvae allows for an improved knowledge of this flora. This information could indicate optimal places to trap individuals, how to improve control methods, and how to locate nests by triangulating the most visited plants.

## Figures and Tables

**Figure 1 animals-13-03038-f001:**
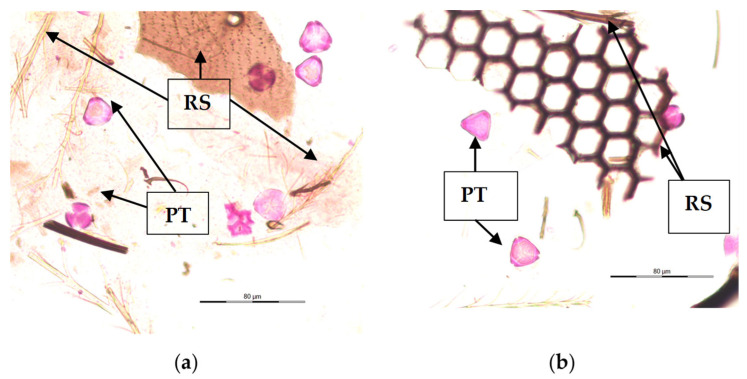
Gastrointestinal contents of *V. velutina* larvae. (**a**,**b**): both images describe the main elements identified in the gastrointestinal contents. PT: Pollen types; RS: remains of structures of insects.

**Figure 2 animals-13-03038-f002:**
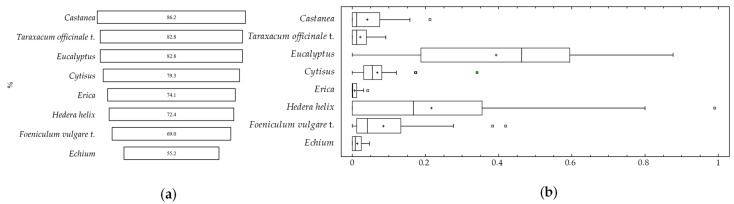
Pollen types present in more than 50% of the samples. (**a**) (%) representativeness in the samples and (**b**) box-and-whisker plots for the pollen types present in more than 50% of the samples. Outside points are values which lie more than 1.5 times the interquartile range above or below the box and are shown as small squares. Far outside points are values which lie more than 3.0 times the interquartile range above or below the box and are shown as small squares with plus signs on them.

**Figure 3 animals-13-03038-f003:**
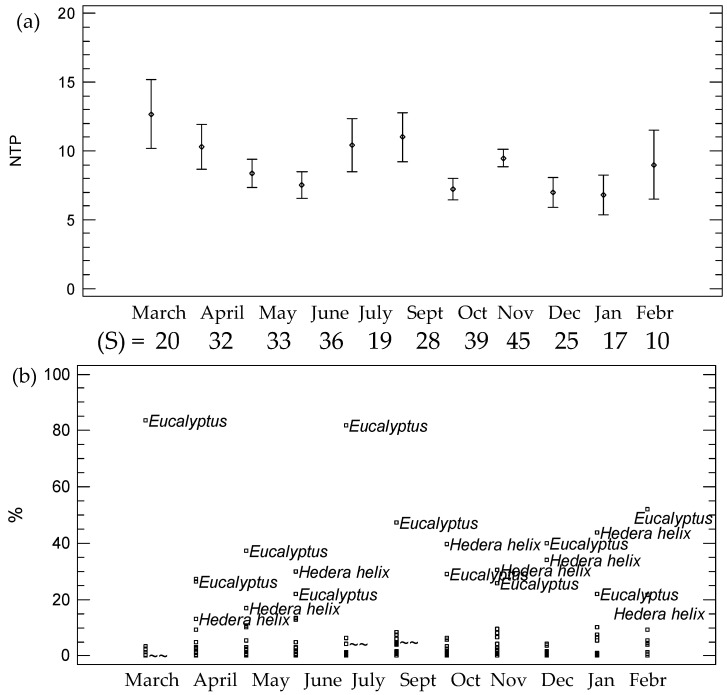
Pollen richness and diversity in the larvae according to the month. (**a**) Fisher’s least significant difference (LSD) procedure for the number of pollen types and (**b**) % of pollen types according to the month. (S) = diversity or number of different taxa according to the month

**Figure 4 animals-13-03038-f004:**
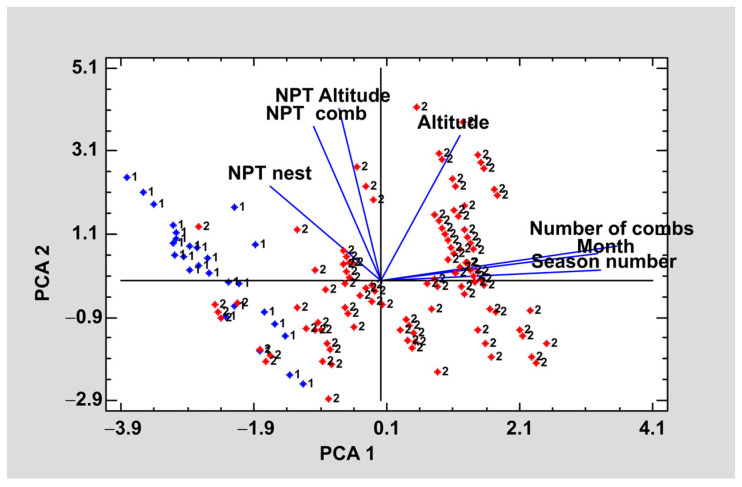
Graphic representation of the first two components of the principal components analysis of the pollen content.

**Table 1 animals-13-03038-t001:** Nests studied: geographical origin, period, type, and number of combs analyzed.

Year	Altitude Range	Altitude	Season	Nest Type	N Combs	N Nest
2016	≤100	30	Winter	Secondary	10	2
2017	≤100	30	Spring	Embryo	3	2
		48	Autumn	Secondary	8	6
	100–200	144	Summer	Secondary	7	1
		475	Autumn	Secondary	7	2
	200–300	143	Autumn	Secondary	10	1
	>300	250	Autumn	Secondary	9	2
2018	100–200	140	Autumn	Secondary	11	1
		140	Winter	Secondary	7	1
2019	≤100	92	Spring	Embryo	1	3
		92	Summer	Secondary	3	2
	100–200	143	Autumn	Secondary	3	1
		141	Winter	Secondary	7	5
2020	≤100	30	Winter	Secondary	12	1
	200–300	250	Autumn	Secondary	9	2
2021	<100	24	Spring	Embryo	1	10
		35		Secondary	3	4
		28	Summer	Secondary	5	1
	100–200	375	Spring	Embryo	1	3
		114		Secondary	3	1
	200–300	143	Spring	Embryo	1	3
	>300	254	Spring	Embryo	1	2

N: Number of.

**Table 2 animals-13-03038-t002:** Main pollen family and pollen types identified in the gastrointestinal contents of *V. velutina.*

Family	Pollen Type	Family	Pollen Type
Amaryllidaceae	*Amaryllidaceae*	Fabaceae	*Fabaceae*
Apiaceae	*Foeniculum vulgare* t.		*Lotus* t.
Aquifoliaceae	*Ilex aquifolium*		*Trifolium* t.
Araliaceae	*Hedera helix*	Fagaceae	*Castanea*
Asphodelaceae	*Asphodelus*		*Quercus*
Asteraceae	*Bellis*	Hypericaceae	*Hypericum*
	*Achillea* t.	Lamiaceae	*Lavandula*
	*Ambrosia* t.		*Mentha*
	*Anthemis* t.		*Thymus*
	*Artemisia* t.	Lauraceae	*Laurus nobilis*
	*Calendula* t.	Lythraceae	*Lythrum*
	*Cardus* t.	Malvaceae	*Malva*
	*Centaurea* t.	Myrtaceae	*Callistemon*
	*Sonchus* t.		*Eucalyptus*
	*Taraxacum officinale* t.	Oleaceae	*Ligustrum*
Betulaceae	*Alnus glutinosa*		*Olea europaea*
	*Betula*	Papaveraceae	*Papaver*
Boraginaceae	*Echium*	Pinaceae	*Pinus*
	*Lithodora*	Plantaginaceae	*Plantago*
	*Pentaglottis*	Platanaceae	*Platanus*
Brassicaceae	*Brassica* t.	Poaceae	*Poaceae*
	*Raphanus* t.	Polygonaceae	*Rumex*
Campanulaceae	*Campanula* t.	Ranunculaceae	*Ranunculus*
Caprifoliaceae	*Abelia*	Rosaceae	*Eriobotrya japonica*
	*Lonicera*		*Prunus*
Caryophyllaceae	*Silene* t.		*Rubus*
Chenopodiaceae	*Chenopodiaceae*		*Crataegus* t.
Cistaceae	*Cistus*	Rutaceae	*Citrus*
Crassulaceae	*Sedum*	Salicaceae	*Salix*
Ericaceae	*Arbutus*	Saxifragaceae	*Saxifraga* t.
	*Calluna*	Scrophulariaceae	*Buddleja davidii*
	*Erica*	Simaroubaceae	*Ailanthus*
Fabaceae	*Acacia*	Theaceae	*Camellia*
	*Cytisus*	Thymelaeaceae	*Daphne gnidium*

t.: Pollen type.

**Table 3 animals-13-03038-t003:** Pollen richness and diversity of pollen according to the nest type and altitude.

Nest Type	Altitude Range	(NT)	Maximum	Minimum	(S)
Embryo nest (S) = 47 (NT) = 9.4	<100	10.2	17	3	46
	100–200	9.7	11	8	14
	200–300	5.7	10	4	13
	>300	9.5	11	8	12
Secondary nest (S) = 67 (NT) = 8.2	<100	7.2	15	2	54
	100–200	8.4	14	3	25
	200–300	9.5	16	4	43
	>300	10.8	22	4	37

(S) = diversity or number of different taxa; (NT) = average content of pollen types.

**Table 4 animals-13-03038-t004:** Pollen richness and diversity of pollen according to the number of combs.

	Comb 1	Comb 2	Comb 3	Comb 4	Comb 5	Comb 6	Comb 7	Comb 8	Comb 9	Comb 10	Comb 11
(S)	51	43	40	38	30	37	36	30	31	21	13
Average	8.9	7.9	8.2	8.1	7.6	9.3	8.8	8.8	9.4	8.0	8.5
Minimum	3	4	3	3	3	4	2	3	6	2	8
Maximum	17	13	15	16	11	14	15	22	13	12	9

(S) = diversity or number of different taxa.

## Data Availability

The datasets that were generated in and/or analyzed in the current study are available from the corresponding author on reasonable request.

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
