# Peer review of "Describing the Pollen Content in the Gastrointestinal Tract of Vespa velutina Larvae"

_animals, 2023, doi:10.3390/ani13193038_

Round 1

Reviewer 1 Report

The manuscript establishes that there are significant amounts of pollen in the guts of larval Vespa velutina but fails to show if this has any nutritional significance for the wasps.  It puts great emphasis on diversity of pollen in the larval guts but does not clearly indicate why we might expect any patterns in this.  The paper could easily be trimmed down.  The report of pollen in the guts of larval Vespa could be interesting but it would be much more so if they can indicate the pollen is of nutritional value and not just a byproduct of preying on pollen laden bees which are chewed into a mush and fed to the wasp larvae.  Hopefully the pollen slides have been retained and they can be checked for signs of digestion of the pollen contents.

While intelligible, the English has many problems and needs a thorough review.  I would not think it publishable as is.

Author Response

Comments on Animals #

This paper does exactly what its title says it does, describe the pollen content in the guts of Vespa velutina larvae. Unfortunately, the manuscript has many problems. The English is very poor and not to the standards of a respectable international journal. This needs a thorough review and revision.

First of all, thank the reviewer for the correction task.

A major revision of the English language has been carried out to improve the understanding.

Regarding the manuscript, we understand that a description of the pollen content from the larvae of Vespa velutina provides information about the habitat where the nests are located and at least the pollen resources used by the prey. We study 675 larval samples collected over 6 years, all of which contained pollen grains. Unfortunately, we cannot confirm that these pollen grains play a significant role in the hornet nutrition, but further research can clarify this role. Most of the pollen grains we have seen and identified were empty and at least partially digested, corresponding to the description made by:

Babendreier, D., Kalberer, N., Romeis, J., Fluri, P., & Bigler, F. (2004). Pollen consumption in honey bee larvae: a step forward in the risk assessment of transgenic plants. Apidologie35(3), 293-300.

However, this does not mean that pollen grains are digested to be used as a nutrient, because the influence of digestive enzymes in the midgut can contribute to this appearance. In fact, it would be very interesting to deep in this hypothesis but more studies in this line should be done. Our first explanation is that the pollen grains pollen go into the hornet gut because it is present above the captured insects. Another question is whether pollen grains are really important for nutrition and, so that Vespa velutina should prefer to capture the insects that carry pollen rather than those that do not, which a priori does not seem to be observed. Otherwise, we observed the foraging behaviour of V. velutina in flowers taking nectar or looking for insects.

Our study highlights the presence of pollen grains in the gut larvae and the type of pollen grains commonly found. As you can see in Figure 3 (b), Eucalyptus is the most abundant pollen grain, mainly for two reasons: it is one of the main melliferous plants of the coastal areas, where the invasive species is well-settled, and it has a long flowering period.

In any case, we understand that the manuscript must be thoroughly revised.

The authors state the study was undertaken in Galicia, Spain but give no specifics. I am not familiar with Galicia beyond tourist literature but I doubt it is ecologically homogeneous. Actual locales for where samples were taken should be given along with some ecological information.

We have incorporated a description of the origin of the nests and a section about the study area.

The authors present a standard life history for V. velutina with overwintering queens, founding of a new colonies in the spring, colony growth and then the eventual production of future reproductives followed by colony collapse and abandonment in late fall yet somehow they have larval gut samples year round. This doesn't add up. A more thorough accounting of the life cycle of V. velutina in Galicia is needed.

Although it is true that autumn is described as the beginning of the decline of the nests, the end of the cycle occurs when the conditions do not allow the flight of the individuals being possible to observe alive individuals, even at the end of January. This nests despite being active, will be abandoned. The next queens (for the following cycle) have been seen starting February or even at the end of January, depending on the meteorological conditions. This is the reason for having nests for all over the year.

In the text it is described as follows:

“This stage lasts until when the unfavorable environmental conditions limit the flight of the workers. The nest is then gradually depopulated and finally it is completely abandoned. In the meantime, the new queens choose a protected place for overwintering.”

However, text corrections have been made for better understanding.

Altitude (presumably of the site where a sample was taken) is used as a major variable in some of the analyses yet no rationale is given for this and the altitudinal differences seem very small.

Yes, the altitude is related to the sampling site. The altitude of the nests ranged between 2 and 600 m above sea level. This aspect has been included in the text.

We include altitude as an important factor because it determines, for example the date of the first observation of the invasive species and the date for initiating nest construction, but also because there is a strong relationship between altitude and vegetation type. Of course, many factors affect vegetation distribution, but at lower altitudes Eucalyptus globulus starts the melliferous production (this means since the end of October to April and even May); at higher altitudes other species (namely Castanea is the main melliferous species) however Eucalyptus camaldulensis can appear.

Altitude can also be related to the size of the nests; lower altitudes should allow for larger sizes because the season start earlier. The larger the nest, the greater the number of combs and the more larvae it contains. In any case, at lower altitudes in Galicia, greater floristic diversity was appreciated. However, this study shows few differences in pollen content, which could indicate how V. velutina can obtain plant resources and invade high altitude areas. This justification has also been added to the text.

The authors clearly demonstrate the presence of pollen in the guts of larval V. velutina but do not show if it has any ecological significance. At a minimum they need to indicate if the pollen, or more properly the pollen cytoplasm, was digested. The pollen exine (what was stained with fuchsin stain) is inedible for most organisms so the cytoplasm within must be somehow extracted to be of any nutritional value. Finding many lysed or otherwise empty pollen grains would indicate possible digestion but this is not mentioned. The authors make no effort to indicate if the amounts of pollen in the gut were nutritionally significant.

Although pollen with degraded exine was observed as discussed in the manuscript, the aim of this study is not to demonstrate the nutritional value of pollen, but rather to describe which species are of interest for V. velutina. As mentioned above, most of the identified pollen grains seem to be digested, but the meaning of this digestion is not clear. In any case, some comments on this are included (Third paragraph of the discussion section).

The values given, number of grains per 10 microliters of solute, are meaningless for that purpose. One can extrapolate those values to estimate the numbers of grains per gut and come up with values ranging between 7500 and 150000 pollen grains per gut. This may seem like a lot but for bees, most of which depend entirely on pollen as a protein source, even 150000 grains would be a tiny fraction of the pollen required to make a bee the size of a large wasp like V. velutina. It would seem unlikely that pollen is playing a major dietary role in the life history of V. velutina, and if they aren't digesting it, no role at all. Of course, the precise importance values depend on pollen grain volume but most of the pollens here seem of average size, nothing super large or small so counts may be sufficient.

Regarding the value of pollen grains/10μL, it was included for the purpose of comparison between samples, not to give absolute values regarding the content in the gut.

According to the comment, we agree that the pollen content in the gut of bees is comparatively much higher, at least this is what we have observed, in honey bee samples that we have also studied for other purposes. But the hornets are very active predators using the structures rich in proteins such as muscles of meat balls for feeding larvae, being this the main protein resource. So, as you mention, even if pollen really contributes to nutrition, it would never be quantitatively significant. However, it would be interesting to define what the qualitative contribution is, for example in terms of providing essential amino acids. Again, further research would help to explain this.

It is unfortunate the authors made no attempt to determine if their populations of V. velutina were actively collecting pollen. Ueno (2015) did look at V. velutina foraging at flowers and stated they did not actively collect pollen. If so, most, if not all of the pollen will be coming from prey such as honey bee workers which V. velutina notoriously capture as they return to the hive, often pollen laden. The authors acknowledge this but then they seem to ignore that any variation in pollen diversity through the season would be entirely mediated by what the bees are collecting (or feeding on in the nest), not choices by the wasps, unless they think the wasps are picking and choosing which bees (or other prey) they kill based on the pollen they carry.

This has been discussed before. We don’t see hornets collecting pollen and they do not have special structure to do it. But some pollen grains are gathered over their body when they visited flowers. Our opinion is that the pollen grains inside the hornet comes both from pollen contamination carried by pollinators (especially bees) and from the consumption of flower nectar. This statement comes from the authors' own observations. Therefore, the pollen is indicative of the plant species with which this hornet interacts. In times of consumption of bees, of course, it is mainly marked by the pollen that the bees carry, but bees carry the resources of the area where they forage. The authors are confident that 675 larvae analyzed over six years throughout the life cycle can contribute to outline a pattern of plants that serve as a foraging site for different insects, including the hornet.

The authors also claim that most insects are pollinators, which of course is not true.

This statement has been amended to read as follows:

“In addition, it is important to highlight that most of the insects hunted by V. velutina contribute to pollination, thus having an impact on this biological process [19].”

I do not understand fig. 2a. I believe Castanea has a somewhat restricted flowering period (3 months?) so how can Castanea pollen be showing up in 86% of the samples unless some is coming from something like honey bee storage. A similar problem appears in Fig. 3.

In our area Castanea starts the blooming period since May (the earliest period) to August in the latest one. Effectively it appears in many samples, but it is not the most abundant, being this Eucalyptus.

Castanea can appear in larvae due to it is present in flying insects that are preyed but also it can be appear as suggested by the reviewer due to their presence inside honeybee colonies. This species is commonly one of the most relevant floral resources as the harvest season for honey bees progresses. At lower altitudes Castanea is one of the last melliferous plants in flowering, so that the nectar and the pollen are stored inside the honey bee colonies until harvest. At higher altitudes, Castanea is crucial for honey production and honey bee colonies maintenance.

It is in autumn (September-November) when the hornets exert the greatest predation pressure on the apiaries, frequently causing the collapse of hives. The hornets can get inside the hive capturing the bees, the larvae (I there are) and eat the honey reserves.

In any case, it is also worth noting that the larger nests have more combs compared to those of the period when Castanea did not yet flower, this help to explain also the fact that a large number of samples contain this pollen grain.

Table 4 seems to be about diversity of pollen types per comb, not the number of combs. If not that should be clarified but I found this whole section very confusing. It seems like diversity here is just the number of pollen types yet this is somehow different from the number of pollen types.

Pollen diversity is the number of different pollen types found, while the number of pollen types is the number of each found. They are different data. For example, in the total number of combs, 51 species (or different pollen types) were differentiated, but in terms of the quantity in each comb there was an average of 8.9 per comb. We have rewritten the text for better understanding.

Also it would have been useful to show if these numbers are related to the total amount of pollen grains per individual and also to have used some measures of evenness, an important measure of diversity, rather than just number of kinds. Although, as mentioned previously, it not clear if any of this is under the direct control of the wasps.

We have ruled out this relationship between the NTP and the amount of pollen/10uL since a high amount does not necessarily imply a high NTP. For example, eucalyptus pollen may be present in the entire sample in high amounts.

The discussion is full of redundancies, rehashing statements repeated several times earlier in the paper. These should be eliminated.

Ok, the discussion was revised and changed avoiding repetitions.

Reviewer 2 Report

This manuscript reports on an original idea and finding associated with and important invasive insect. However I question whether this phenomenon has any ecological meaning. Instead I believe it to be a study on a random curious interaction that cannot be used to infer any significant ecological interactions, although the manuscript repeatedly says that it does.

Firstly, the Asian hornet is a predator and derives it proteins from insects, spiders and scavenged vertebrates. The authors do not provide any evidence that the pollen grains can contribute to the nutritional status of the larvae being fed in the nest. In fact the pollen is just a by-product of a forager catching an insect with pollen on it, or more rarely when feeding on nectar resources. One thus expects the pollen found in the larva’s gut to be any pollen grain found in the foraging range of the colony (super organism). I do hovever also note that many of these nests were in trees with pollen that was found in the larvae (e.g., Eucalyptus species) suggesting there could also be passive pollen transfer to workers leaving and entering the nest.

Second the varying diversity in pollen types over time and the range of plant species that are documented from the guts of larvae of a nest are not particularly interesting as it is well known that pollen availability in the environment varies over space and time.

In the discussion there is a whole paragraph that is irrelevant to the study, although it is used as rationale for the investigation, namely line 295-300: “Nectar and pollen are nutritionally superior to other plant materials and are a main food source for many bees and wasps. Like other hymenopteran bees and wasps, hornets require carbohydrates as a key energy source; floral nectar is one of the major sugar resources for them. In fact, hornets are frequently observed visiting flowers to feed on nectar [17,19-20]. Plant resources are especially valuable in adults as source of energy for activities such as flight [21]. However, as regards the feeding of the larvae, it is not known.” It is irrelevant that other bees and wasps derive nourishment from pollen because V. velutina is a predatory wasp and uses animal proteins instead of plant proteins. You have to provide evidence that pollen is a food resource to predatory social wasps before you can make this argument. Otherwise it is just speculation.

Line 350 to 352: “However, a high pollen diversity has been observed during the months of October and November, which could mean that the hornets have had to seek and provide a greater variety of plants/ or protein resources to have the same nutritional contributions than previous month.” This sentence sounds like you are saying that the foragers are selecting more diverse pollen sources, but this would not be possible as the pollen they carry is simply a sample of what is available. The wasp nest cannot receive any nutritional benefit from the pollen being fed to the larvae. Rephrase to make clear what is meant.

Line 383-385: “These results provide information on the flora surrounding the habitat of this hornet and on the feeding behavior of V. velutina. This could be used to improve control and surveillance methods.” This concluding sentence is weak and practically I cannot see any benefit to knowing the pollen profile in a wasp nest to improve control and surveillance. It is also a circular argument because you would need to sample a wasp nest to know what pollen is in the gut. Mapping nests and relating it to plant species occurrence in a radius around the nest would be much less work that analysing the pollen.

Minor comments:

First sentence of the simple summary the saying is used incorrectly – ‘turned’ not ‘brought’

The rationale given in the Abstract is also misleading to the non-specialist reader because, i) the pollen is not actively foraged for and ii) there is no proof that nutritional benefit is derived from ingesting pollen.

Line 354: “The location of the nest conditions the choice of resources used.” – Meaning not clear.

Minor editing of English language required

Author Response

Reviewer 2

First, the authors would like to thank the reviewer for his time and comments. Below, we will address each of the points raised by the reviewer and respond to the comments made point by point.

This manuscript reports on an original idea and finding associated with and important invasive insect. However I question whether this phenomenon has any ecological meaning. Instead I believe it to be a study on a random curious interaction that cannot be used to infer any significant ecological interactions, although the manuscript repeatedly says that it does.

This study is an approach to the diet of V. velutina larvae. We understand that it may seem random, however, the number of samples studied (675) over 6 years provides a description of the flora that surrounds its ecosystem and could imply a pattern of feeding behavior in the areas studied.

Firstly, the Asian hornet is a predator and derives it proteins from insects, spiders and scavenged vertebrates. The authors do not provide any evidence that the pollen grains can contribute to the nutritional status of the larvae being fed in the nest. In fact the pollen is just a by-product of a forager catching an insect with pollen on it, or more rarely when feeding on nectar resources. One thus expects the pollen found in the larva’s gut to be any pollen grain found in the foraging range of the colony (super organism). I do hovever also note that many of these nests were in trees with pollen that was found in the larvae (e.g., Eucalyptus species) suggesting there could also be passive pollen transfer to workers leaving and entering the nest.

Since 2016 the authors have detected how this hornet can feed on a highly diverse diet. Although as we mentioned in the text, V. velutina especially uses other insects and arachnids to provide proteins for its larvae, it also uses (own observations) sugars directly from plants, especially those from which it can obtain various resources (pollen, insects, and nectar). The objective of the article was not to determine the nutritional contribution of this pollen, but rather to describe the pollen of the plants that is associated with this hornet either for feeding, or as a place to hunt or reproduce.

Although it is true that some of this pollen may appear as secondary contamination, all this information guides us towards the knowledge of the flora of interest of V. velutina and the ecosystem that surrounds it, which could help to delimit its location and help in control methods.

However, the text has been corrected to make this objective clearer.

Second the varying diversity in pollen types over time and the range of plant species that are documented from the guts of larvae of a nest are not particularly interesting as it is well known that pollen availability in the environment varies over space and time.

Pollen diversity is variable; however, the study area has a greater diversity and not all of this has been seen in the gastrointestinal content. There is a flora of interest to this hornet. In the same way that honey bees use those plants that provide them with nectar and/or pollen, V. velutina is more related to some species than to others.

The text has been corrected to make this result clearer.

In the discussion there is a whole paragraph that is irrelevant to the study, although it is used as rationale for the investigation, namely line 295-300: “Nectar and pollen are nutritionally superior to other plant materials and are a main food source for many bees and wasps. Like other hymenopteran bees and wasps, hornets require carbohydrates as a key energy source; floral nectar is one of the major sugar resources for them. In fact, hornets are frequently observed visiting flowers to feed on nectar [17,19-20]. Plant resources are especially valuable in adults as source of energy for activities such as flight [21]. However, as regards the feeding of the larvae, it is not known.” It is irrelevant that other bees and wasps derive nourishment from pollen because V. velutina is a predatory wasp and uses animal proteins instead of plant proteins. You have to provide evidence that pollen is a food resource to predatory social wasps before you can make this argument. Otherwise it is just speculation.

We agree with the reviewer. We have modified the text based on this correction.

Line 350 to 352: “However, a high pollen diversity has been observed during the months of October and November, which could mean that the hornets have had to seek and provide a greater variety of plants/ or protein resources to have the same nutritional contributions than previous month.” This sentence sounds like you are saying that the foragers are selecting more diverse pollen sources, but this would not be possible as the pollen they carry is simply a sample of what is available. The wasp nest cannot receive any nutritional benefit from the pollen being fed to the larvae. Rephrase to make clear what is meant.

The authors mean that a greater number of plants were visited for foraging, either to consume carbohydrates or to hunt other insects. The text has been rephrased to clarify the meaning.

Line 383-385: “These results provide information on the flora surrounding the habitat of this hornet and on the feeding behavior of V. velutina. This could be used to improve control and surveillance methods.” This concluding sentence is weak and practically I cannot see any benefit to knowing the pollen profile in a wasp nest to improve control and surveillance. It is also a circular argument because you would need to sample a wasp nest to know what pollen is in the gut. Mapping nests and relating it to plant species occurrence in a radius around the nest would be much less work that analysing the pollen.

We have modified the text to make the conclusion more understandable.

Although we provide an explanation below:

  1. velutina is a hornet with a wide range of resources to meet its nutritional needs. In fact, its most relevant impacts include beekeeping, the fruit sector, loss of pollinator diversity. Its high reproductive capacity has shown that they are a species that adapts very well to a variety of ecosystems. It is the flora around the nest that provides them with most of these resources. However, they do not use all the flora in their environment equally. Some plant species are more attractive than others. Although it is easy to see individuals flying on certain plants, it is difficult to monitor all the plants visited by this hornet. Identifying the pollen in its gastrointestinal contents of larvae could provide information on which flora V. velutina visits to prey on insects or to acquire carbohydrates. Knowing this flora could indicate places to trap individuals, but also to locate nests by triangulating from the most visited plants.

Minor comments:

First sentence of the simple summary the saying is used incorrectly – ‘turned’ not ‘brought’

Done

The rationale given in the Abstract is also misleading to the non-specialist reader because, i) the pollen is not actively foraged for and ii) there is no proof that nutritional benefit is derived from ingesting pollen.

The abstract has been modified based on this suggestion.

Line 354: “The location of the nest conditions the choice of resources used.” – Meaning not clear.

The sentence has been changed to “The location of the nest determines the choice of resources used.”

Comments on the Quality of English Language

Minor editing of English language required

We appreciate your corrections and suggestions. Finally, we have checked all the text and corrected the English of the manuscript.
